# A Comparative Investigation of the Surface Properties of Corn-Starch-Microfibrillated Cellulose Composite Films

**DOI:** 10.3390/ma16093320

**Published:** 2023-04-23

**Authors:** Zuzanna Żołek-Tryznowska, Ewa Bednarczyk, Mariusz Tryznowski, Tomasz Kobiela

**Affiliations:** 1Faculty of Mechanical and Industrial Engineering, Warsaw University of Technology, Narbutta 85, 02-524 Warsaw, Poland; ewa.bednarczyk@pw.edu.pl (E.B.); mariusz.tryznowski@pw.edu.pl (M.T.); 2Faculty of Chemistry, Warsaw University of Technology, Noakowskiego 3, 00-662 Warsaw, Poland

**Keywords:** corn starch, starch-based films, surface, surface free energy, microfibrilated cellulose, MFC

## Abstract

Starch-based materials seem to be an excellent alternative for conventional plastics used in various applications. Microfibralted cellulose can be used to improve the surface properties of starch-based materials. This study aims to analyze the surface properties of starch-microfibrillated cellulose materials. The surface properties of films were evaluated by ATR-FTIR, surface roughness, water wettability, and surface free energy. The surface homogeneity between corn starch and microfibrillated cellulose (MFC) fibers was confirmed by scanning electron microscopy (SEM) and atomic force microscopy (AFM). Microscopic analyses of the film surfaces confirm good compatibility of starch and MFC. The addition of MFC increased the surface roughness and polarity of developed starch/MFC materials. The surface roughness parameter has increased from 1.44 ± 0.59 to 2.32 ± 1.13 for pure starch-based materials and starch/MFC material with the highest MFC content. The WCA contact angle has decreased from 70.3 ± 2.4 to 39.1 ± 1.0°, while the surface free energy is 46.2 ± 3.4 to 66.2 ± 1.5 mJ·m^−2^, respectively. The findings of this study present that surface structure starch/MFC films exhibit homogeneity, which would be helpful in the application of MFC/starch materials for biodegradable packaging purposes.

## 1. Introduction

Plastics are excellent packaging materials, but their most significant disadvantage is how their waste is managed [1,2,3]. Plastic packaging does not degrade in the natural environment or degrades over hundreds of years, contributing to pollution and creating the so-called microplastics [4,5]. The impact of these on the environment and human life has not yet been thoroughly researched [6,7]. Plastic film packaging for food is often used as single-use packaging—it becomes a waste after use. Global plastic waste production reached approximately 353 million tons in 2019, and 40% of plastic waste comes from packaging [8]. At the same time, only 9% of plastic waste produced globally is recycled. Due to environmental protection, relatively high oil prices, and exhausting sources of fossil fuels, conventional plastics will be slowly replaced by plastics from renewable raw materials [9,10].

Starch-based materials appear to be a viable alternative to plastic materials [11,12]. Starch is a polysaccharide. It is readily available, biodegradable, cheap, and renewable [13]. It can be an excellent natural raw material for obtaining biodegradable and environmentally friendly packaging materials. Furthermore, starch can be harvested without destroying the plant [14]. Starch-based films are isotropic, odorless, tasteless, colorless, non-toxic, and biodegradable [15,16]. Starch films will only partially replace conventional plastic packaging, but they can be a great alternative in some applications, for example, food packaging. The ideal starch-based film packaging can be used for products with a short shelf life, and after usage, they can be biodegraded in home composting containers. The addition of plant extracts or plant essential oils can increase their antibacterial or antioxidant properties resulting in smart packaging [17]. Various modifications of starch can be performed to improve their final material properties. For example, UV-B irradiation can be successfully applied to change the properties of corn and potato starches [18]. Pająk et al. reported that the use of honey-bee products improves the mechanical properties of potato starch edible films [19]. Da Fonseca de Albuquerque et al. studied the influence of cold plasma influence on the water absorption and wettability of corn starch film surface [20]. Furthermore, various blends can be used to obtain films for packaging purposes. Ramakrishnan et al. developed films for packaging obtained from alginate/carboxymethyl cellulose/starch ternary blends [21]. The use of ternary blends allowed them to obtain materials with improved mechanical properties.

Microfibrillated cellulose, MFC, is an excellent material that can be utilized to reinforce the properties of starch films. MFC is composed of high-aspect-ratio nanosized fibrils in the 10–100 nm range. It is characterized by a high specific surface area and a large number of surface hydroxyl groups which can generate inter- and intramolecular hydrogen bonds [22,23]. Briefly, MFC is obtained by mechanical or chemical destruction of the multi-level organization of natural fibers in cellulose and then extraction of the microfibrils [24,25]. A recent literature survey showed that using MFC materials has great potential. MFC can be successfully used as a filler for hydrophobic biopolymers such as PLA [26]. This is due to the capability of MFC to form a rigid and homogenous network characterized by low porosity. Fotie et al. reported using MFC as a coating for food packaging plastics, whereas changes in the oxygen barrier properties were observed [27]. Furthermore, MFC can be successfully used for the development of coatings for paper packaging [22]. Wei et al. investigated starch-corn stalk cellulose material for biodegradable straws [28] instead of plastic straws, which are restricted in UE. MFC is a suitable material for piezoelectric composite films’ matrix phase and is intended for flexible wearable electronic sensors [29].

Various microscope techniques are successfully used to observe the microstructure of starch granules [30] as well as plasticized films developed by casting techniques, e.g., [31,32,33]. Atomic force microscopy, AFM, is a powerful tool that can be used to study a film surface and provide quantitative and qualitative information about the compounds of the film on a nanometric scale [34]. Thiré et al. compared glycerol-plasticized and non-plasticized cast film morphology [35]. The AFM technique allowed the observation of smooth and rough regions on the surface of the developed starch films. Using the AFM technique, Dai et al. compared the surface morphology of starch films based on starches from different botanical sources and other commercially modified cassava starches [36]. According to Dai et al., the AFM technique allows the observation of at least three types of morphology surface structures depending on the type of starch used for the starch film development [36].

This research aims to evaluate the addition of microfibrillated cellulose on the surface properties of corn-starch films. To the best of our knowledge, only a few papers deal with the surface properties of MFC/starch materials. The knowledge of surface properties might be crucial for applications such as printing or coating. In this work, we depend on the knowledge of the surface properties of starch/MFC composites. The surface roughness was measured with an optical microscope, and the surface properties of the films were confirmed by SEM and AFM micrographs. Then, the surface’s wettability was assessed, and the values of the surface free energy and its components, polar and dispersion, were determined.

## 2. Materials and Methods

### 2.1. Materials

Starch from corn (CAS 9005-25-8, Prod. No. S4126, moisture content ≤ 15%) and diiodomethane (CAS 75-11-6, Prod. No. 158429, purity ≥ 98.5%) was purchased from Sigma-Aldrich (Poznań, Poland). Microfibrilated cellulose, MFC, was a water-based suspension with a fiber diameter of 2 nm–10 μm and lengths > 200 μm. Distilled water was used for the starch solution. Water purified by electrodeionization with a Millipore Sigma Elix Water Purification System was used for the contact angle measurement. All chemical reagents were used as received.

### 2.2. Preparation of Starch/MFC Composites

The detailed preparation method can be found in our previous work, i.e., see [37].

Briefly, the starch, glycerol, and MFC solution in water was heated up to 95 °C at a rate of 3 °C per min while continuously mixing at a speed set at 500 rpm using a mechanical stirrer. Next, the mixture was cooled down to approx. 50 °C and poured on Teflon^®^ plates placed on a K Paint Applicator (RK Print, Royston, UK) equipped with an adjustable micrometer spreader gap set at 3 mm with a constant coating speed (6 m·min^−1^). The investigated films were prepared twice from the freshly prepared solution. The MFC content was 0%, 0.2%, 0.6%, 1%, 2%, 3%, and 4% *w*/*w* relative to the starch dry basis. After drying the composites in the climate chamber (23 ± 0.5 °C, 50 ± 1 RH), the films were gently peeled out from the Teflon^®^ plates by hand. The compositions of the starch/MFC materials are presented in Table 1.

All of the measurements were conducted one week after film preparation. This method ensures obtaining repeatable films. Properties measuring the quality of the films were visually assessed.

### 2.3. Infrared Spectroscopy (ATR FT-IR)

ATR FT-IR spectra were recorded on the films for each composite formulation, at room temperature, in the 400–4000 cm^–1^ range with a resolution of 4 cm^−1^, using a Nicolet iS5 spectrometer (Thermo Scientific, Madison, WI, USA) equipped with a platinum single-reflection diamond ATR module. The FT-IR spectra were analyzed with OMNIC Specta™ software (series 9.12.968).

### 2.4. Surface Morphology

A digital microscope (Keyence VHX7000, Keyence Corporation, Osaka, Japan) equipped with a VH-Z100R objective was used for the microscopic analysis of the linear film roughness arithmetical mean height R_a_. Three-dimensional images of the film surfaces were taken and analyzed with the Keyence VHX-7000. The surface roughness parameter was calculated as a mean from six roughness profiles.

The developed films’ morphological characteristics were assessed using a JOEL JCM-7000 (Jeol Ltd., Tokyo, Japan) scanning electron microscope, SEM, with an acceleration voltage of 15 kV. The samples of films were covered with a thin layer of gold prior to SEM image determination.

Sample surfaces were analyzed using an atomic force microscope, AFM (Park Systems XE-120), operating in the contact mode. At least two tips (PPP-CONTSCR, Nanosensors) for each sample were used to ensure reproducibility. All scans were performed at room temperature. For each sample, several images were taken at different scan sizes and various places to gain better knowledge of variations in the local structures. For all images, measurements were started with the same values of scan parameters. The scan rate was 1 Hz, and the set point was 1.3 nN. However, in each case, final optimizations were performed. AFM data were analyzed by XEI, an Image Processing Program for SPM data developed by Park Systems. From AFM measurements, it is possible to calculate the parameters which characterize surface roughness [38]. The following surface parameters were considered: wettability measurement and surface free energy determination.

The water contact angle of the obtained films was measured using a DSA 30E drop shape analysis system (Krüss, Germany) according to the ISO 15989 standard. The sessile drops on the film surface were deposited with a 0.5-diameter needle. The drop shape analysis was completed using Advance software. The reported values are the mean of six probes on two various films. The SFE was calculated using the Owens–Wendt–Rabel–Kaelbe approach.

### 2.5. Statistical Analyses

If possible, the data were expressed as mean ± SD (standard deviation). Statistical analyses were evaluated using Statgraphics Centurion 19 (v.19.1.3 StatPoint^®^, Inc., Warrenton, VA, USA) software. One-way variance analysis (ANOVA) was used for the data analyses, and the averages were compared by the Tukey test with a significance level of 95% (*p* < 0.05).

## 3. Results and Discussion

The starch/MFC composites were obtained in the form of a transparent film. All of the films were transparent and clear with a smooth undersurface, which had contact with the Teflon^®^ plate, and a more matted upper surface. In this work, we focused on the surface characterization of the starch/MFC composites obtained.

The ATR-FTIR analyses allow revealing the surface functional groups. Figure 1 shows the ATR-FTIR spectra of pure starch film, MFC after drying, developed films, and the difference between the ATR-FTIR spectra before and after 30 days. The broadband around 3300 cm^–1^ corresponds to O—H stretching vibrations related to intermolecular and intramolecular hydrogen bonds of hydrogen groups of starch [39] and MFC [40]. An enhanced peak follows at 1630 cm^–1^. It is from the H—O—H bending vibration of bound water in both materials [41]. This water acts like a plasticizer. The characteristic peaks for C—O bond stretching are attributed to the 895, 989, 1071, and 1144 cm^–1^ peaks, whereas 989 and 1107 cm^–1^ correspond to the anhydroglucose O—C stretching. The sharp peak at approx. 2900 cm^–1^ is characteristic of C—H_2_ stretching in starch and MFC molecules. The FTIR analysis suggested the formation of hydrogen bonding between starch and MFC particles without creating chemical bonds between them.

Figure 1a shows the FTIR spectra of films with increasing content of MFC. All of these spectra have a similar profile because all of the films have identical components. Due to the similarity of the functional groups in the starch and MFC molecules, there is no typical band that allows for differentiation of both compounds and observing the changes of starch-MFC films with increasing the content of MFC.

Regarding the ATR-FTIR results after 30 days (see Figure 1b), the analyses revealed a broadening of the peak at 3300 cm^–1^, which is typical for water. This means that more hydroxyl groups of water are involved in the formation of intra- and intermolecular hydrogen bonds within those materials. Hence, we can conclude that the absorption of the water on the surface of the starch film is possible due to the presence of hydroxyl groups in the starch and MFC molecules. The films did not change their appearance during that time.

The 3D microscopic photos allowed us to determine the roughness profiles of the developed films. Table 2 summarizes the basic roughness parameters R_a_, which describe the geometry and irregularity of the film’s surface. The images of the film surface observed by the optical microscope are shown in Figure 2 and Appendix A. The roughness surface, R_a_, defines the arithmetic mean deviation of the height of a line. The basic roughness parameter describes surface microgeometry and relating to specific profile features [42].

Generally, an increase in the MFC content of the starch-based films increases the roughness, R_a_, of the developed films.

This might be related to the good interactions and miscibility between the starch granules and microcellulose fibers. The MFC fibers can act as a plasticizer in the starch film matrix, like glycerol acts as a plasticizer [43]. The images taken for the roughness measurements show a lack of surface deformation or cracks. A naked human eye can also see the changes in the rough surfaces when less matting and greater surface shining are visible. Chen et al. reported an increase in the surface roughness with the increase in the MFC in the MFC-starch-PVA composite films [44], confirming the increase in pores on the film surface when the content of MFC greater than 10%. Furthermore, according to Liu et al., an increase in nanoparticles in the film increases the surface roughness of films due to the protuberances in the starch films [45,46].

To confirm this phenomenon, SEM micrographs were taken. Scanning electron microscopy (SEM) was used to examine the microstructural view of the surface (see Figure 3a,b and Appendix A). Micrographs of the surface were taken at a magnification of 1000× and did not reveal any differences among the films with increasing content of MFC. All the films showed smooth surfaces without cracks, breaks, or pores, indicating good miscibility, compatibility, and integrity of starch and MFC. Furthermore, the micrographs of developed films did not reveal the aggregation of MFC fibers in the material. This is also related to strong hydrogen bonding between MFC fibers and starch granules [47].

AFM has been used to observe the morphology of the starch films. Figure 4 shows the topographic images of developed films. The AFM topographic images confirmed the absence of residual starch and MFC granules or fibers, respectively, showing that all the components are compatible during the solution casting process. The topography obtained by the AFM technique confirms that the films are relatively smooth and homogenous, without any pores or cracks, and exhibit good structural integrity. AFM topographic images of developed starch-MFC films revealed insignificant differences in morphology structures with the increasing content of MFC. The films are flat with small particles having peaks up to but not exceeding 3 μm. Furthermore, the values of RMS and Ra determined by the AFM technique confirm the changes in the surface roughness determined by the optical microscope. Topographic images of films at higher magnifications revealed smooth domains, confirming the observation with optical and SEM microscopes.

The water contact angle (WCA) is the most common method for determining materials’ wettability or surface hydrophobicity. The value of WCA indicates whether the surface is hydrophobic (WCA > 90°) or hydrophilic (WCA < 90°) [48]. In general, the starch-based films are hydrophilic and show an affinity for water due to the presence of hydrogen bonds. On the other hand, a more hydrophobic surface and less water absorbance are desired for packaging usage. The Owens–Wendt–Rabel–Kaelbe (OWRK) approach was used for the surface free energy SFE calculation [49]. Table 3 shows the values of water contact angle followed by SFE and its components values; Figure 5 shows the WCA and SFE and polar component changes.

The WCA gives valuable information about developed films’ wettability and hydrophilic character. All produced MFC/starch-based films are hydrophilic, characterized by values of WCA lower than 90°. The results showed that WCA values ranged from 70.3 ± 2.4° to 39.1 ± 1.0° for S_0 and S_4 films, respectively. All developed MFC/starch-based films are hydrophilic, characterized by values of WCA lower than 90°. The variation in WCA might be related to hydroxyl bonds from starch and MFC and simultaneously with the higher surface roughness of the S_4 film. The values of WCA for the starch-based films are similar to those reported in the literature [50]. The ANOVA test results for the significance of MFC content in the starch films on the WCA reveal a *p*-value lower than 0.05, so the influence of the MFC content is statistically significant.

In general, the surface free energy and the polar component of SFE of developed films increase when the content of MFC increases. The starch film S_0 exhibited 46.2 ± 3.4 mJ·m^−2^ and 7.8 ± 2.8 mJ·m^−2^ for SFE and polar SFE components, respectively. On the other hand, the starch film with higher content of MFC exhibited the highest values of the SFE and polar SFE components, 66.2 ± 1.5 and 22.2 ± 1.1 mJ·m^−2^, respectively. This might be related to the surface structure of MFC-starch films and the higher content of hydroxyl groups on the material’s surface.

## 4. Conclusions

In this work, we developed starch/MFC composites to analyze their surface properties. Both raw materials have good film-forming properties and exhibit good compatibility due to inter- and intramolecular hydrogen bonding. The FTIR analysis suggested the formation of hydrogen bonding between starch and MFC particles. Our results reveal that increased MFC content in the starch-MFC films increases the surface roughness. SEM and AFM micrographs confirmed a homogeneous structure and the absence of cracks and pores. The surface polarity increased with the increasing content of MFC. This might be related to the higher content of hydroxyl groups on the material surface. MFC can be successfully used to improve the properties of starch films, i.e., roughness and wettability.

## Figures and Tables

**Figure 1 materials-16-03320-f001:**
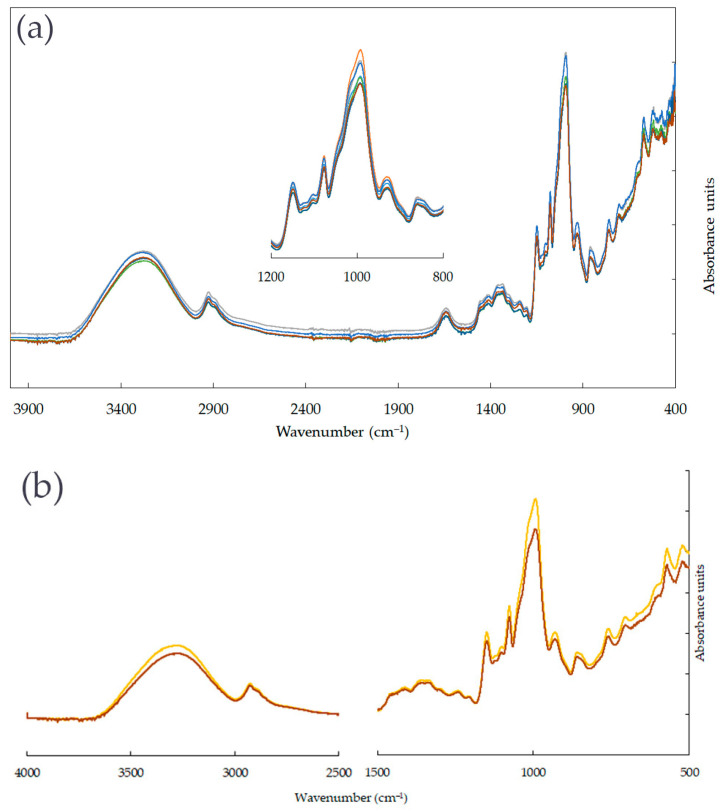
FTIR spectra of (**a**) starch films and pure MFC, (**b**) films S_3 and S_3 before and after 30 days.

**Figure 2 materials-16-03320-f002:**
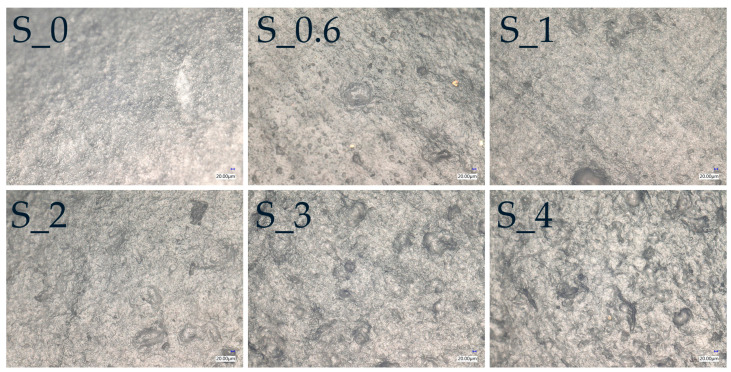
Images of the developed film surface taken by optical microscopic at magnification 200×.

**Figure 3 materials-16-03320-f003:**
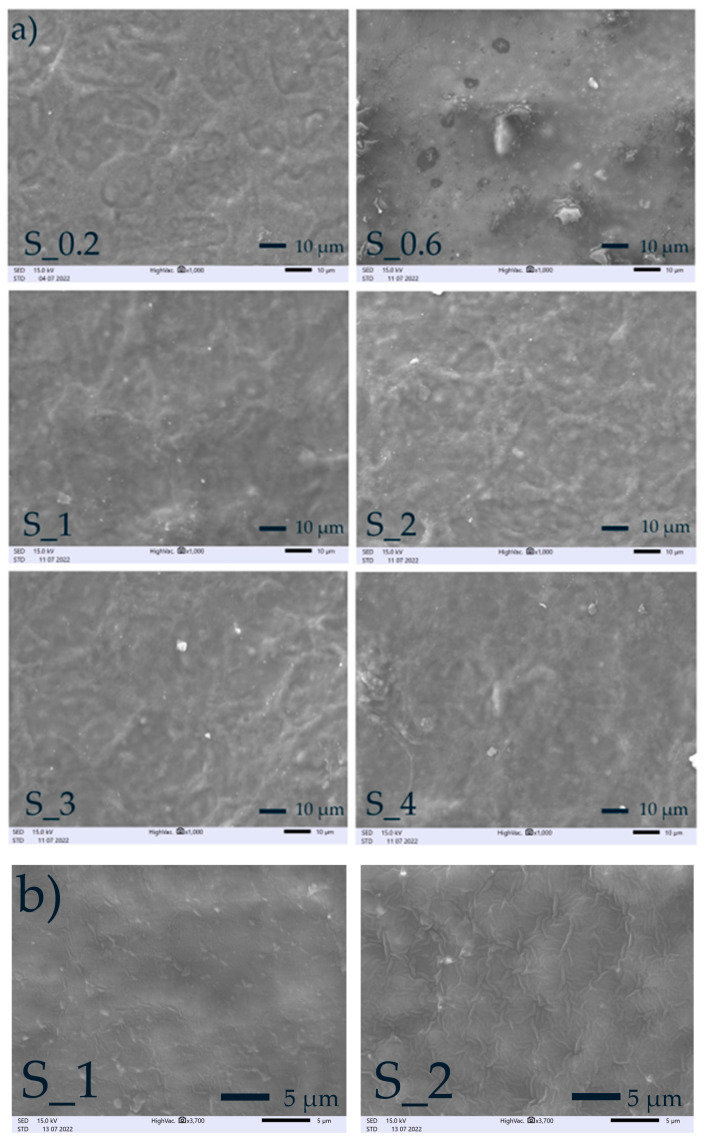
SEM photos of films: (**a**) micrographs at magnifications 1000×; (**b**) micrographs at magnifications 3700×.

**Figure 4 materials-16-03320-f004:**
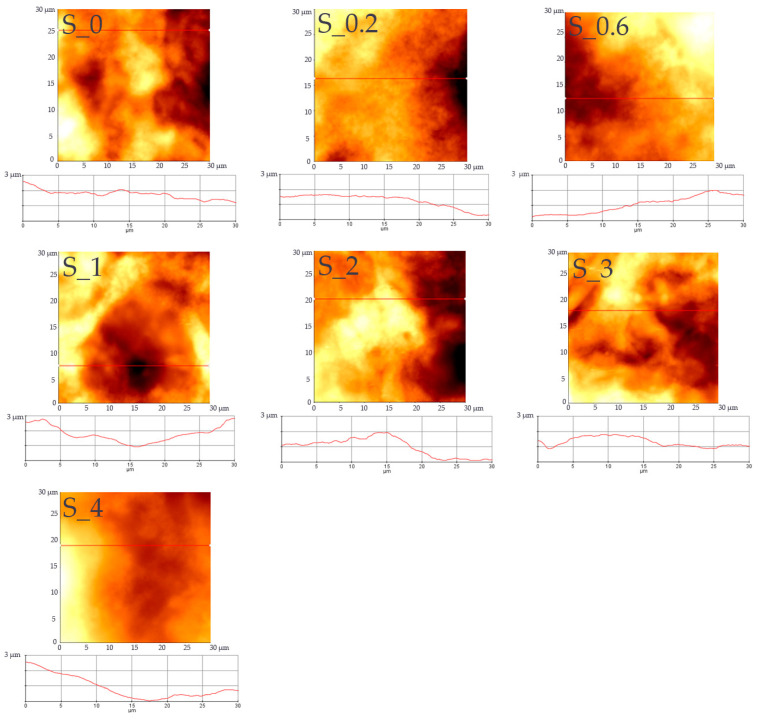
AFM topographic images of developed films.

**Figure 5 materials-16-03320-f005:**
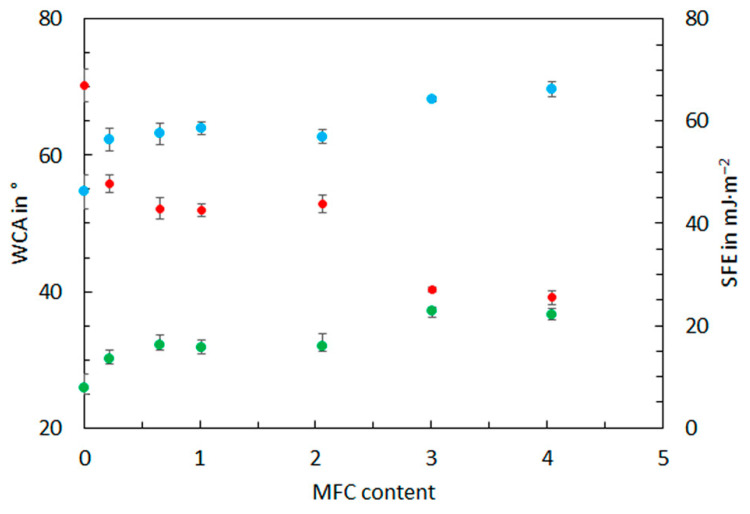
**Figure 5**. Changes in the WCA (red dots), SFE (blue dots), and polar component (green dots) with MFC content.

**Table 1 materials-16-03320-t001:** Formulations of starch-based composites.

Run	Film Abbreviation	Starch (g)	MFC (g)
1	S_0	10.00	−
2	S_0.2	10.04	0.22
3	S_0.6	10.02	0.66
4	S_1	10.04	1.02
5	S_2	10.06	2.06
6	S_3	10.06	3.01
7	S_4	10.02	4.04

**Table 2 materials-16-03320-t002:** Roughness profiles, R_a_, of the film’s surface.

Film	R_a_ in μm
S_0	1.44 ± 0.59 ^ab^
S_0.2	2.07 ± 1.05 ^abc^
S_0.6	2.89 ± 1.27 ^a^
S_1	1.14 ± 0.42 ^a^
S_2	2.89 ± 0.59 ^bc^
S_3	2.52 ± 1.19 ^abc^
S_4	2.32 ± 1.13 ^c^

Values are means ± SD, *n* = 6 per treatment group. Means having the same letter for a parameter are not significantly different (*p* < 0.05) through the Tukey test.

**Table 3 materials-16-03320-t003:** The values of water contact angle and surface free energy components.

Film	WCA * (°)	Surface Free Energy (mJ·m^−2^)		
		Polar	Dispersive	Total
S_0	70.3 ± 2.4 ^c^	7.8 ± 2.8 ^a^	38.3 ± 2.1 ^a^	46.2 ± 3.4 ^a^
S_0.2	55.9 ± 1.3 ^b^	13.5 ± 1.6 ^b^	42.8 ± 1.5 ^bc^	56.4 ± 2.2 ^b^
S_0.6	52.2 ± 1.5 ^b^	16.2 ± 2.0 ^b^	41.3 ± 1.0 ^b^	57.5° ± 2.2 ^b^
S_1	51.9 ± 0.9 ^b^	15.7 ± 1.4 ^b^	42.9 ± 0.9 ^bc^	58.6 ± 1.2 ^b^
S_2	52.8 ± 1.3 ^b^	16.0 ± 2.4 ^b^	41.0 ± 2.0 ^b^	57.0 ± 1.4 ^b^
S_3	40.3 ± 0.4 ^a^	22.8 ± 0.7 ^c^	41.4 ± 0.5 ^b^	64.2 ± 0.5 ^c^
S_4	39.1 ± 1.0 ^a^	22.2 ± 1.1 ^c^	44.0 ± 0.6 ^c^	66.2 ± 1.5 ^c^

* Measured after sessile drop deposition at the first contact. Values are means ± SD, *n* = 6 per treatment group. Means having the same letter for a parameter are not significantly different (*p* < 0.05) through the Tukey test.

## Data Availability

The data presented in this study are available on request from the corresponding author.

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
