# Peer review of "A Comparative Investigation of the Surface Properties of Corn-Starch-Microfibrillated Cellulose Composite Films"

_materials, 2023, doi:10.3390/ma16093320_

Round 1

Reviewer 2 Report

Dear editor

In this study, author have investigated the surface property of micro fibrillated cellulose/Corn starch composite as biocompatible materials using SEM and AFM. The wettability, surface free energy and roughness were also evaluated.

However, authors have to address below comments:

·         More quantitative and numerical results are necessary in the abstract. The results must be quantitavely show the wettability, surface free energy and roughness for improved surface property in comparison. The conclusion of the study must be added as a last sentence of abstract. The preparation method should be mentioned in Abstract.

·         The text requires editing of English language.

·         Revise Figure caption in a scientific manner. For example” Table 1. Formulations of starch-based composites in g”. Replace “in g” with scientific sentence.

·         Start the “Results and discussion” section with in a scientific manner. Starting with “After drying” is not suitable and it hasn’t continuity.

·         Figures are not uniform and doesn’t have unique style and font. Please revise them. Correct the resolution of Fig.1. Scales in images of Table2 are not clear and the measurement method has not been mentioned. Please explain Figure 2 and 3 captions more in details. It needs uniform style of numbering as recommended.

·         In FTIR results, which absorption band show the difference after adding MFC?

·         There is no literature survey and comparison with similar articles in discussion section.

·         There is no enough references for “results and discussion”. It should be enriched.

·         Author should address the novelty and advantages of the study

Best

Round 2

Reviewer 2 Report

.